# Combination of rRT-PCR and Anti-Nucleocapsid/Anti-Spike Antibodies to Characterize Specimens with Very Low Viral SARs-CoV-2 Load: A Real-Life Experience

**DOI:** 10.3390/microorganisms9061263

**Published:** 2021-06-10

**Authors:** Zoe Florou, Meropi Zigra, Philippos Kartalidis, Katerina Tsilipounidaki, Georgia Papadamou, Aikaterini Belia, George C. Fthenakis, Efthymia Petinaki

**Affiliations:** 1Department of Microbiology, University Hospital, University of Thessaly, 41110 Larisa, Greece; zoi_fl@yahoo.gr (Z.F.); maryz30@hotmail.gr (M.Z.); kartalidisphilip@gmail.com (P.K.); tsilipoukat@gmail.com (K.T.); 2Department of Emergency, University Hospital, University of Thessaly, 41110 Larisa, Greece; georgia.papadamou@yahoo.com; 3Department of Emergnecy, Hospital of Karditsa, 43100 Karditsa, Greece; katebelia1975@gmail.com; 4Veterinary Faculty, University of Thessaly, 43100 Karditsa, Greece; gcf@vet.uth.gr

**Keywords:** anti-nucleocapsid antibodies, anti-spike antibodies, COVID-19, diagnosis, false positive, Greece, low viral load, N gene

## Abstract

The objective of the present study was to evaluate the true positivity among people, whose results of initial testing of nasopharyngeal swabs (NPS) showed a very low viral load of severe acute respiratory syndrome coronavirus 2 (SARS-CoV-2). Seventy-seven people detected with low viral loads of SARs-CoV-2 in nasopharyngeal samples (Ct ≥ 35) were enrolled in the study. For this purpose, a second NPS was collected for rRT-PCR (real-time reverse transcription polymerase chain reaction) combined with a pair of serum samples for detection of anti-nucleocapsid (anti-N) and anti-spike (anti-S) antibodies. In 8 people, subsequent examinations indicated an increase in viral loads, thereafter, followed by an increase of anti-N and anti-S antibodies, findings compatible with an early stage of COVID-19 infection. In 9 people, who already had increased anti-N antibodies, subsequent examination showed a decrease or absence of viral load and an increase in antibodies, indicative of a late stage of COVID-19 infection. In 60 people, subsequent examination showed absence of infection (as indicated by absence of viral load and antibodies). We propose that the combination of a second NPS and one serum-specimen, both taken three days after the first NPS, helps significantly to avoid false-positive results.

## 1. Introduction

SARS-CoV-2 is a novel coronavirus that has emerged in the last year, leading to a worldwide pandemic of COVID-19 disease, that has affected over 219 countries [1,2]. In order to control this new pathogen, the development of novel diagnostics and antiviral therapies is being considered the highest priority. Whole genome sequencing of SARS-CoV-2 led scientists to apply novel molecular approaches in virus-detection and drug-development. During the early period of the pandemic, complete sequencing of SARS-CoV-2 facilitated the specific primer-designing and laboratory diagnosis of COVID-19 [3]. New efforts in novel strategies, employing short peptides, proteins and natural resources, such as plant derivatives, are new ways for disabling the virion at the level of the spike protein, which is a promising target for the treatment of COVID-19 [4]. Current studies are focused on comprehensive characterization of the structure of the spike protein as a crucial step to find new therapeutics, which will interrupt the process of recognition and the entry of virus into the cell [5].

With the continuous spread of COVID-19 at a tremendous rate, the most important challenge faced by the global health community is the rapid and early detection of SARS-CoV-2 infected people. The rapid identification of people infected by this new virus can help to initiate and establish appropriate therapy early, as well as to protect public health as much as possible. Many diagnostic assays have been introduced into clinical practice, which either target viral genes and viral proteins or specific antibodies produced by the infected host [6]. However, the gold standard for identifying SARS-CoV-2 infection relies on the detection of viral RNA by rRT-PCR-based (reverse transcription polymerase chain reaction) techniques [6]; whereas, the results are expressed with a cycle threshold (Ct) value, which is defined as the earliest the fluorescent signal is detectable above the threshold.

In many studies, this Ct value is used as a viral load indicator, despite the fact that it is dependent on several factors, which include the clinical sample quality, the type of the specimen, the period from collection to testing, the sample transportation conditions and the target gene [6]. The Centres for Disease Control and Prevention (CDCP) and the World Health Organization (WHO) have provided guidelines on the interpretation of results with threshold and Ct values; a clinical sample is considered positive when all the SARS-CoV-2 target genes have a Ct value < 40 [7,8,9]. Ιn a systematic review, Rao et al. [10] have indicated that lower Ct values are potentially associated with worse outcomes in people with COVID-19 [10]. In a more recent study, Magleby et al. [11] indicated that samples with Ct < 25 should be considered to have high viral load, those with Ct between 25 and 30 as with medium viral load and samples with Ct > 30 as with low viral load [11].

Although several studies have focused on false-negative RT-PCR results, some recent papers have focused on false-positive results, which might have led to misclassifying people as infected with SARS-Cov-2 [12,13]. The Department of Microbiology of the University Hospital of Larissa (University of Thessaly) is the regional reference laboratory for the diagnosis of SARS-CoV-2 infection in Central Greece, an area with a population of approximately 1,000,000 people. Since the start of the COVID-19 pandemic, the department has been receiving clinical samples from five local hospitals for molecular detection of the virus.

During that period, a significant diagnostic issue arose with the interpretation of rRT-PCR (real-time reverse transcription polymerase chain reaction) results, in which Ct ranged between 35 and 39 in the official reporting of such findings. It was also noted that a brief evaluation of the clinical history of people with low viral load (Ct ≥ 35) indicated that around half of these people did not show any clinical signs compatible with COVID-19, but rather had been tested for various monitoring reasons (e.g., general population monitoring, contact with a person with confirmed infection, pre-surgical evaluation in hospitals etc.).

The objective of the present study was to evaluate the true positivity among people whose results of initial testing of nasopharyngeal swabs showed a very low viral load of SARS-CoV-2. For this purpose, a second nasopharyngeal sample was taken for rRT-PCR combined with a pair of serum specimens for the detection of anti-nucleocapsid (anti-N) and anti-spike (anti-S) antibodies.

## 2. Materials and Methods

### 2.1. Organization of Work Flow SARS-CoV-2 Detection by Reverse Transcription Real Time PCR (rRT-PCR)

Nasopharyngeal samples (NPS) taken from people at any of the five hospitals of Central Greece were submitted for detection of SARs-CoV-2 and were processed immediately. Viral RNA was extracted from 400 μL from each NPS by using the commercial kit MagDEA^®®^Dx SV using a magLEAD^®®^ 12gC instrument (Precision System Science Co, Matsudo city, Chiba, Japan). Detection of SARS-CoV-2 was performed by rRT-PCR, by using a commercial kit that targeted the *E* (common for other SARS-related coronaviruses) and *N* (specific for SARS-CoV-2) genes (Direct SARS-CoV-2 Real-Time PCR kit, Vircell, Granada, Spain), with a threshold limit of detection of 3.5 copies per reaction for both genes. The RNase *P* gene region was used as an endogenous internal control for the analysis of biological samples (Direct SARS-CoV-2 Real-Time PCR kit, Vircell, Granada, Spain).

A sample was considered to be SARS-CoV-2 positive, when Ct values for both the *E* and *N* genes were found to be <40, according to the recommendations of the manufacturer. In addition, samples in which Ct values for the RNAse *P* gene were found to be ≥40, were rejected.

After testing, all the NPS and RNAs were kept at −20 °C and −80 °C, respectively.

A copy of the final result of the test for each sampled person was sent to the referring clinicians, who were responsible for informing the people. The result of the test was also added into the Greek national platform e-Government Center for Social Security (IDIKA), as required by the national policy on the measures against COVID-19.

### 2.2. Selection and Enrollment of People into the Study

People, in NPS from whom the rRT-PCR yielded a Ct value for the *N* gene ≥35, but <40, were informed of the result by the clinicians and were considered for enrollment into the study, if they also fulfilled the below criteria:No previous diagnosis of COVID-19 (clinical or laboratory findings).No previous laboratory test found positive for SARS-CoV-2 (molecular or immunological test).No history of recent contact with a person with confirmed (by laboratory testing) COVID-19.

Before inclusion into the study, all people were informed of the details of the study and expressed their willingness to participate in it. Demographic and medical details for each participant (age, gender, reason for taking the initial NPS, etc.) were obtained by the clinicians.

### 2.3. Second Nasopharyngeal Sampling

Three days after collection of the initial NPS (NPS-1), a second NPS (NPS-2) was collected from each person enrolled in the study and was submitted for laboratory testing. NPS-2 was also processed as described above by means of rRT-PCR.

At the same time, viral RNA from the NPS-1 of the same person was re-extracted and re-tested by rRT-PCR (2nd run NPS-1) along and under the same conditions as NPS-2, as described above.

### 2.4. Detection of Anti-Nucleocapsid and Anti-Spike Antibodies

On the occasion of obtaining NPS-2, a blood sample (BS-1) was collected from each person enrolled in the study. A second blood sample (BS-2) was subsequently collected two weeks after BS-1.

Serum was prepared from the blood samples for antibody detection. Detection of anti-nucleocapsid (anti-N) and anti-spike (anti-S) IgG antibodies was performed using the commercial assays Elecsys^®®^ Anti-SARS-CoV-2 and Elecsys^®®^ Anti-SARS-CoV-2 S, respectively, in a cobas e 602 module (Roche, Basel, Switzerland).

With regard to anti-N antibodies, values for the ratio S/Co < 1 were considered to be negative and values ≥ 1 were considered to be positive. With regard to anti-S antibodies, values < 0.8 μL^−1^ were considered to be negative and values ≥ 0.8 μL^−1^ were considered to be positive.

### 2.5. Data Management and Analysis

Based on the combination of the results of rRT-PCR and the antibody titers, three groups of people were created retrospectively, as below.

In group A, were allocated people, with the following results: (a) decrease of Ct for the *N* gene from 2nd run NPS-1 to NPS-2 and (b) increase of anti-N antibody titers from BS-1 (negative) to BS-2 (positive).In group B, were allocated people, with the following results: (a) increase of Ct for the *N* gene from 2nd run NPS-1 to NPS-2 to ≥40 and (b) further increase of anti-N antibody titers from BS-1 (positive) to BS-2 (positive).In group C, were allocated people, with the following results: (a) Ct for the *N* gene (a1) ≥ 40 at 2nd run NPS-1 or (a2) > 35 at 2nd run NPS-1 and ≥40 at NPS-2 and (b) no increase of anti-N antibody titers from BS-1 (negative) to BS-2 (negative).

During evaluation of Ct for the rRT-PCR, values were rounded to the nearest unit (half unit values (i.e., *.5) were rounded to the higher unit). For the analysis, Ct values for the *N* gene were used, as previous studies have indicated that these had a higher specificity compared to Ct values for the *E* gene [14]. For the purposes of the statistical analysis only, in rRT-PCR, Ct values found to be ≥40, were given the arithmetic value of 40 to facilitate the computations; as this was the lowest possible negative value, statistical differences were not increased artificially.

The change in Ct values between NPS-1 and NPS-2 was calculated as the result of subtraction of Ct value obtained during the 2nd run NPS-1 from the Ct value obtained during the NPS-2 of rRT-PCR for the *N* gene. The change in antibody titers was calculated as the result of subtraction of the titers obtained for BS-1 from that obtained for BS-2; separate changes were calculated for anti-N and anti-S antibodies.

Data were entered into Microsoft Excel and analyzed using SPSS v. 25 (IBM Analytics, Armonk, NY, USA). Basic descriptive analysis was performed.

Frequencies were compared by using cross-tabulation with Pearson’s chi-square test or Fisher exact test, as appropriate.

Ct values for the *N* gene obtained from 1st run NPS-1 and 2nd run NPS-1 were compared between them by using the Wilcoxon signed-rank test. The same test was also used to compare Ct values obtained in 2nd run NPS-1 and NPS-2, as well as anti-N and anti-S antibody titers obtained in BS-1 and BS-2. Differences between groups were evaluated by means of the Kruskal-Wallis test.

Correlation analysis was performed between changes in Ct values and antibody titers.

The potential association of changes in Ct values with changes in antibody titers was assessed by analysis of correlation. Correlations and correlation coefficients are those of Pearson.

In all analyses, statistical significance was defined at *p* < 0.05.

## 3. Results

### 3.1. Total Number of Samples Processed for Diagnosis of SARS-CoV-2 Infection

From March 2020 until February 2021, a total of 78,150 NPS, collected from 28,760 different people, were tested by rRT-PCR in our department for the diagnosis of SARS-CoV-2 infection among the population in Central Greece.

A total of 11,320 samples (14.5% of total specimens) had Ct values for the N gene, which ranged from 11 to 39. Specifically, 10,130 samples (12.9% of total specimens) had Ct values between 11 and 34 and were reported as “positive”, whilst the other 1190 samples (1.6% of total specimens) had Ct values between 35 and 39 and were reported as “presumed positive”.

### 3.2. Enrolment of People into the Study and Allocation to Groups

In total, 77 people (median age: 51 years, min.–max.: 35–70 years) met the criteria for inclusion and were willing to participate, thus, they were enrolled into the study, from October 2020 to January 2021. Of these, 37 were female (51, 35–70 years) and 40 were male (53, 35–70 years).

Cumulative results of the Ct values of rRT-PCR for the *N* gene in 1st run-NPS-1 (i.e., before their inclusion in the study) are in Table 1. Ct values of rRT-PCR for the RNase *P* gene in 1st run-NPS-1 were always <22.

Based on the combined results of laboratory tests, of the 77 people 8 were retrospectively allocated to group A, 9 to group B and 60 to group C. There were not significant differences in the gender (*p* = 0.49), nor in the median age (*p* = 0.062) between people allocated to each of the three groups. However, all 8 people with clinical signs compatible with COVID-19 (e.g., fever, malaise, respiratory signs), were allocated to group A or B (*p* < 0.0001). In addition, there were no differences in the median (min.–max.) Ct value of rRT-PCR for the *N* gene between the three groups at 1st run NPS-1 (group A: 36 (35–36), group B: 35 (35–36), group C: 36 (35–39); (*p* = 0.16)).

### 3.3. Results of rRT-PCR for the N Gene in NPS from People into the Study

There was significant difference between the Ct values of 1st run NPS-1 and 2nd run NPS-1 of rRT-PCR for the *N* gene: the median values (min.-max.) were 36 (35–39) and 40 (33–40), respectively (*p* < 0.0001). Ct values of 1st run NPS-1 of rRT-PCR for the *N* gene indicated that all people (100%) had a SARS-CoV-2 infection, whilst Ct values of 2nd run NPS-1 indicated that only 22 people (28.6%) had a SARS-CoV-2 infection (*p* < 0.0001).

There was significant difference between the Ct values of 2nd run NPS-1 and NPS-2 for rRT-PCR for the *N* gene for people in group A or B. In group A, a decrease was noted: the median value of Ct decreased from 34 (33–35) to 28 (22–28) (*p* = 0.01); in contrast, in group B, an increase was evident: from 35 (35–37) to 40 (40–40) (*p* < 0.01). No such difference was seen for the results of people in group C and median values of Ct were the same: 40 (37–40) and 40 (40–40) (*p* > 0.05) (Figure 1).

### 3.4. Results of Antibody Titers in Blood Samples from People in the Study

In group A, titers were low at BS-1 and increased at BS-2 for both anti-N and anti-S antibodies: from 0.08 (0.08–0.08) to 8.80 (5.30–11.80) and 1.65 (0.08–4.20), respectively (*p* < 0.01 for both types of antibodies). In group B, titers were already high at BS-1 and increased further at BS-2 for both anti-N and anti-S antibodies: from 12.40 (7.57–22.30) to 25.30 (16.40–54.00) and from 0.08 (0.08–8.60) to 12.40 (6.70–23.04), respectively (*p* < 0.01 for both types of antibodies). No such changes were evident in samples from people in group C, in whom antibodies were always 0.08 (for all samples and both sampling occasions) (Figure 2).

### 3.5. Associations between Results of rRT-PCR in NPS and Results of Antibody Titers in Blood Samples from People into the Study

Among results of people in groups A and B, there was a clear correlation between the change in Ct values and the change in both anti-N (*r* = 0.4243, *p* = 0.045) and anti-S (*r* = 0.8232, *p* < 0.0001) antibody titers (Figure 3). The difference between the two correlation coefficients was also significant (*z* = 1.89, *p* = 0.029). No such correlation could be calculated for results of people in group C.

### 3.6. Differentiation of People as ‘True Positive’ or ‘True Negative’ for COVID-19

The combination of the results of RT-PCR with the titers of anti-N and anti-S antibodies allows the characterization of people into Group C as ‘true negative’ for COVID-19. If any people of this group had an early-stage infection, which was failed to be detected by rRT-PCR (Ct > 40), an increase of antibodies would be observed within the next two weeks. On the other hand, people in group A are characterized as ‘true positive’ for COVID-19; the increase of viral load in-between the two samples and the absence of antibodies in the first serum are in concordance with an early stage COVID-19 infection. Finally, people in group B are also ‘true positive’ for COVID-19; the detection of antibodies in the first serum sample, combined with the decrease of the viral load in-between the two samples, is indicative of a late-stage COVID-19 infection.

## 4. Discussion

Nucleic acid amplification testing is still considered to be the gold standard for the diagnosis of SARS-CoV-2 in clinical specimens. However, the problem of inaccurate results of rRT-PCR (false positive and false negative) is being increasingly reported [15,16]. To avoid false positive or negative results, each clinical setting must define its Ct cutoff value, taking under consideration the limit of detection of the assay and the distribution of Cts in the population being tested [17]. In a recent study by Falasca et al. [18], 1% of samples tested (17 of 1639) were found to contain a very low viral RNA load, since positivity was detected at high Ct. In the same study, a second NPS collected 24 h after the first, indicated that 12 of 17 samples were negative (70.5%). Laboratory data correlated well with clinical data further supported these laboratory findings, as only 2 among these 17 persons developed clinical signs relevant to COVID-19 [18]. Katz et al. [19] reported false-positive rRT-PCR reactions in a setting for urgent head and neck surgery and otolaryngologic emergencies, leading to delay of urgent surgical procedures and transfer of people to COVID-19-designated units [19].

In our study, the results of rRT-PCR coupled with findings regarding anti-N and anti-S antibodies were used to correctly characterize 77 cases which had a very low viral load. It is estimated that the highest viral load (Ct ranging from 14 to 24) occurs on the third day post-illness onset; when samples would be collected before or after this timepoint, lower viral load can be detected, showed as increased Ct values (Ct > 35) [16]. The collection of a second NPS, three days after the initial ‘presumptive’ positive NPS, can provide further information regarding the kinetics of viral load. Samples from people at the initial stage of the disease would show an increase of viral load; in contrast, samples from patients during the convalescence stage of the infection would show decreasing viral load, or could even show negativity, three days later.

Decreasing viral load might also occur in true negative cases, i.e., non-infected people, of whom the first sample was a false-positive, possibly due to contamination or high test sensitivity. This can be more often the case in people with no history of clinical signs relevant to COVID-19, as shown in the current study, findings that are in line with those of Falasca et al. [18].

Unfortunately, at the moment, no guidelines or studies are available regarding management of such cases (i.e., with Ct > 35). The World Health Organization has recommended the collection of a second NPS (although the precise time of collection is not mentioned), when the result of RT-PCR testing is negative, but there is continuing suspicion of COVID-19 on clinical grounds; if the result of the second NPS is also negative, paired blood samples, with an interval of 2 to 4 weeks, should be collected [6].

Although the detection of a low amount of viral RNA is crucial for the identification of a positive individual, the interpretation of high Ct values (i.e., ones denoting low viral load) should be handled with caution. Based on the analysis of the findings of the present study, we believe that the combination of collecting a repeat NPS and a blood sample three days after the initial NPS, would help substantially to avoid false-positive results. We also postulate that, possibly, the second blood sample might not have been necessary and only created a delay in the management of people. As antibodies are detected one week after development of clinical signs, the detection of anti-N antibodies, combined or not with anti-S antibodies, contributes to the differentiation of people who are virus ‘naive’ to those who are in the convalescence period. People with a negative repeat NPS and not detectable antibodies in the serum sample could be considered SARS-CoV-2 ‘negative’ [20,21,22].

## 5. Conclusions

In the diagnosis of COVID-19, false positive results may have several effects, for example, the allocation of resources to people not really needing them, the unnecessary quarantining of people found to be infected, the necessity for tracing of contacts of the person etc. As the pandemic continues, false positives of laboratory diagnostic evaluations should be kept to a minimum, to allow better management of the pandemic situation and avoid unnecessary burden to the national health systems.

In this context, we propose that all cases with a low viral load need, prior to final reporting, further investigation. The combination of a second nasopharyngeal sample and a serum sample, to be collected three days after the initial nasopharyngeal sample would contribute significantly to the accurate diagnosis of COVID-19.

## Figures and Tables

**Figure 1 microorganisms-09-01263-f001:**
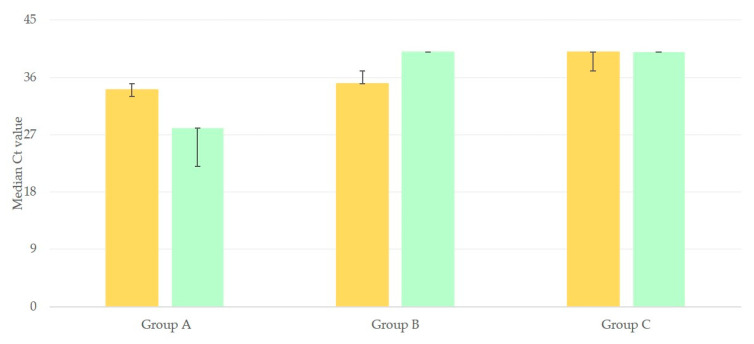
Median Ct values of rRT-PCR for the *N* gene, for nasopharyngeal samples obtained 3 days apart (initial sample: light ochre, second samples: light green) and processed under the same conditions (bars indicate min.–max. values).

**Figure 2 microorganisms-09-01263-f002:**
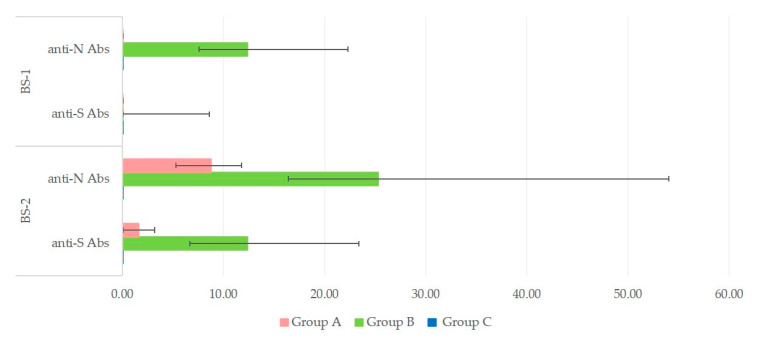
Median values of anti-nucleocapsid (anti-N) and anti-spike (anti-S) antibody titers in blood samples collected 15 days apart, BS-1 and BS-2 (bars indicate min.–max. values).

**Figure 3 microorganisms-09-01263-f003:**
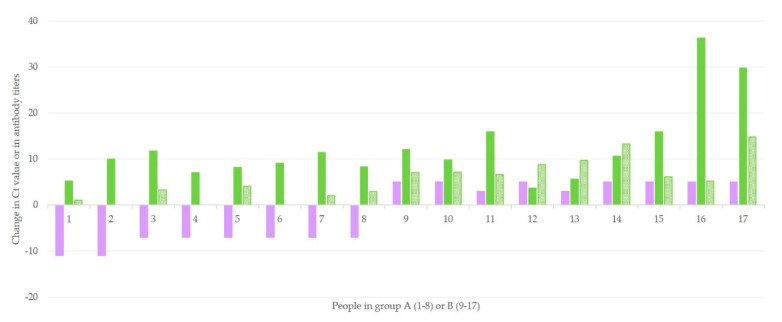
Associations of progressive changes in Ct values between NPS-1 and NPS-2 (obtained 3 days apart) of rRT-PCR for the *N* gene (violet bars) with changes in antibody titers between BS-1 and BS-2 (obtained 15 days apart) for anti-N (green full pattern bars) and anti-S (green massif pattern bars) in 17 patients with SARS-CoV-2 infection.

**Table 1 microorganisms-09-01263-t001:** Cumulative results of rRT-PCR for *N* gene in samples from people (*n* = 77), who were subsequently enrolled into the study.

Ct Value in rRT-PCR for the *N* Gene	No. of People
35	30
36	25
37	14
38	4
38	4
Total	77

## Data Availability

All data associated with the study have been included into the national platform e-Government Center for Social Security (IDIKA), as required by the Greek national policies on the measures against COVID-19.

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
