# Peer review of "Combination of rRT-PCR and Anti-Nucleocapsid/Anti-Spike Antibodies to Characterize Specimens with Very Low Viral SARs-CoV-2 Load: A Real-Life Experience"

_microorganisms, 2021, doi:10.3390/microorganisms9061263_

Round 1

Reviewer 1 Report

This manuscript focuses on the subject of sensitivity of PCR assays for SARS CoV2 infection. The authors developed a simple but effective approach to increase the sensitivity of the PCR assay in people with high Ct values, by doing a subsequent testing 3 days after the initial test. This together with serology for nucleocapsid and spike antibodies help to clearly identify the false positives and false negatives. 

It will be good if the manuscript text is corrected for English language as it will help for easy reading.

Author Response

RESPONSE TO REVIEWERS FOR microorganisms-1207504

Detailed response to comments of the reviewers.

-Reviewer 1

This manuscript focuses on the subject of sensitivity of PCR assays for SARS CoV2 infection. The authors developed a simple but effective approach to increase the sensitivity of the PCR assay in people with high Ct values, by doing a subsequent testing 3 days after the initial test. This together with serology for nucleocapsid and spike antibodies help to clearly identify the false positives and false negatives.

AU:     We thank the reviewer for appreciating our work.

It will be good if the manuscript text is corrected for English language as it will help for easy reading.

AU:     The manuscript has been extensively reviewed for English, as indicated, and various typographical errors have been corrected.

-Reviewer 2

I have a minor comment. Please include the catalog numbers of all the reagents used in the study for the benefits of the readers.

AU: We thank the reviewer for the consideration of our work.             

The precise details of all reagents used have been added, with full names and manufacturers, as indicated.

-Reviewer 3

This is a good paper that presents a new formulation for monitoring the PCR test for detecting COVID-19 by looking in detail at viral loads and timing of the detection procedure. The construction of such testing strategies seem to be microbiologically sound, owing to a mechanism of avoiding false positives.

AU:     We thank the reviewer for appreciating our work.

However, it is possible that such low viral cases might be detected sooner than within the interval of the Author's proposed strategy. Is there a way to be faster than 3 days? I missed an outlook within the discussion and conclusions.

AU:     The issue has been addressed in the Discussion to some extent. Unfortunately, at the moment, there are neither official WHO guidelines, nor other relevant studies. We propose an interval of three days between the two NPS in order to observe a more clear difference of viral loads.

From a viewpoints of referencing and comparing, I missed the proposed method of the Authors state-of-the-art part with respect to former virus disease detection strategies. I do not request the whole COVID-19 literature review, however, the benchmarks on such methods and some comparison with the proposed one.

AU:     A new sub-section was inserted in the Results, where we provide our thoughts regarding the infection status of people in each of the three groups, A, B, C, based on the results of molecular and immunological techniques.

Another point that the Authors should mention are the COVID-19 novel molecular efforts. This may fit within the introduction section, which comes together with several molecular approaches in the detection, drug-design and therapies for COVID-19, like for example identifying the molecular configurations and binding energies of the the spike protein, as it is the case of Moreira et al. Materials 2020, 13(23), 5362 (https://www.mdpi.com/1996-1944/13/23/5362).

AU:     As suggested, new details have been added in the Introduction of the revised manuscript, where recent relevant papers (including the one by Moreira et al. and other in the same lines) have been reviewed.

Reviewer 2 Report

I have a minor comment. Please include the catalog numbers of all the reagents used in the study for the benefits of the readers.

Author Response

RESPONSE TO REVIEWERS FOR microorganisms-1207504

Detailed response to comments of the reviewers.

-Reviewer 1

This manuscript focuses on the subject of sensitivity of PCR assays for SARS CoV2 infection. The authors developed a simple but effective approach to increase the sensitivity of the PCR assay in people with high Ct values, by doing a subsequent testing 3 days after the initial test. This together with serology for nucleocapsid and spike antibodies help to clearly identify the false positives and false negatives.

AU:     We thank the reviewer for appreciating our work.

It will be good if the manuscript text is corrected for English language as it will help for easy reading.

AU:     The manuscript has been extensively reviewed for English, as indicated, and various typographical errors have been corrected.

-Reviewer 2

I have a minor comment. Please include the catalog numbers of all the reagents used in the study for the benefits of the readers.

AU: We thank the reviewer for the consideration of our work.             

The precise details of all reagents used have been added, with full names and manufacturers, as indicated.

-Reviewer 3

This is a good paper that presents a new formulation for monitoring the PCR test for detecting COVID-19 by looking in detail at viral loads and timing of the detection procedure. The construction of such testing strategies seem to be microbiologically sound, owing to a mechanism of avoiding false positives.

AU:     We thank the reviewer for appreciating our work.

However, it is possible that such low viral cases might be detected sooner than within the interval of the Author's proposed strategy. Is there a way to be faster than 3 days? I missed an outlook within the discussion and conclusions.

AU:     The issue has been addressed in the Discussion to some extent. Unfortunately, at the moment, there are neither official WHO guidelines, nor other relevant studies. We propose the interval of the three days between the two NPS in order to observe a more clear difference of viral loads.

From a viewpoints of referencing and comparing, I missed the proposed method of the Authors state-of-the-art part with respect to former virus disease detection strategies. I do not request the whole COVID-19 literature review, however, the benchmarks on such methods and some comparison with the proposed one.

AU:     A new sub-section was inserted in the Results, where we provide our thoughts regarding the infection status of people in each of the three groups, A, B, C, based on the results of molecular and immunological techniques.

Another point that the Authors should mention are the COVID-19 novel molecular efforts. This may fit within the introduction section, which comes together with several molecular approaches in the detection, drug-design and therapies for COVID-19, like for example identifying the molecular configurations and binding energies of the the spike protein, as it is the case of Moreira et al. Materials 2020, 13(23), 5362 (https://www.mdpi.com/1996-1944/13/23/5362).

AU:     As suggested, new details have been added in the Introduction of the revised manuscript, where recent relevant papers (including the one by Moreira et al. and other in the same lines) have been reviewed.

Reviewer 3 Report

This is a good paper that presents a new formulation for monitoring the PCR test for detecting COVID-19 by looking in detail at viral loads and timing of the detection procedure. The construction of such testing strategies seem to be microbiologically sound, owing to a mechanism of avoiding false postiives. However, it is possible that such low viral cases might be detected sooner than within the interval of the Author's proposed strategy. Is there a way to be faster than 3 days? I missed an outlook within the discussion and conclusions. From a viewpoints of referencing and comparing, I missed the proposed method of the Authors state-of-the-art part with respect to former virus disease detection strategies. I do not request the whole COVID-19 literature review, however, the benchmarks on such methods and some comparison with the proposed one. Another point that the Authors should mention are the COVID-19 novel molecular efforts. This may fit within the introduction section, which comes together with several molecular approaches in the detection, drug-design and therapies for COVID-19, like for example identifying the molecular configurations and binding energies of the the spike protein, as it is the case of Moreira et al. Materials 2020, 13(23), 5362 (https://www.mdpi.com/1996-1944/13/23/5362). 

Author Response

(The authors gave the same response as above.)

Round 2

Reviewer 3 Report

I have no further comments to the revised version of the Ms. A minor observation is the use of titres and titers, whereas almost the whole Ms. has been adapted for using titres, Fig. 3 uses titers in the Y-axis text.